# Investigating Machine Learning & Natural Language Processing Techniques Applied for Predicting Depression Disorder from Online Support Forums: A Systematic Literature Review

**Isuri Anuradha Nanomi Arachchige** [1,*] , **Priyadharshany Sandanapitchai** [2] **and Ruvan Weerasinghe** [1]

1   The Language Technology Research Laboratory, University of Colombo School of Computing, #35, Reid Avenue, Colombo 00700, Sri Lanka; arw@ucsc.cmb.ac.lk
2   François-Xavier Bagnoud Center, Rutgers University, 65 Bergen Street, Newark, NJ 07107, USA; ps1009@sn.rutgers.edu
*   Correspondence: isa@ucsc.cmb.ac.lk

**Abstract:** Depression is a common mental health disorder that affects an individual's moods, thought processes and behaviours negatively, and disrupts one's ability to function optimally. In most cases, people with depression try to hide their symptoms and refrain from obtaining professional help due to the stigma related to mental health. The digital footprint we all leave behind, particularly in online support forums, provides a window for clinicians to observe and assess such behaviour in order to make potential mental health diagnoses. Natural language processing (NLP) and Machine learning (ML) techniques are able to bridge the existing gaps in converting language to a machine-understandable format in order to facilitate this. Our objective is to undertake a systematic review of the literature on NLP and ML approaches used for depression identification on Online Support Forums (OSF). A systematic search was performed to identify articles that examined ML and NLP techniques to identify depression disorder from OSF. Articles were selected according to the PRISMA workflow. For the purpose of the review, 29 articles were selected and analysed. From this systematic review, we further analyse which combination of features extracted from NLP and ML techniques are effective and scalable for state-of-the-art Depression Identification. We conclude by addressing some open issues that currently limit real-world implementation of such systems and point to future directions to this end.

**Keywords:** information extraction; machine learning; depression identification; online forum mining





## 1. Introduction

Depression is one of the most common mental disorders and a major contributor to the world's overall burden of disease [1]. By 2020, it is estimated that 264 million people of all ages will have suffered from depression [1]. The effects of depression can be recurrent or long-lasting and impact the person's quality of life. Psychiatric interview is the main method of diagnosing depression in the clinical setting, through which individuals are asked a series of questions about the symptoms, how long they have had them and how the symptoms have affected their day-to-day life [2]. Accordingly, depression is categorised as mild, moderate and severe. Based on the Diagnostic and Statistical Manual of Mental Disorders (DSM-V) diagnostic criteria, an individual must be experiencing at least five symptoms such as depressed mood, loss of interest in activities that they previously enjoyed, loss of appetite, insomnia or sleeping too much, fatigue, feeling worthless, difficulty concentrating or recurrent suicidal thoughts among others, during the same 2-week period [3]. Standard measures such as Beck's Depression Inventory (BDI) [4], DSM-V [5], International Classification of Diseases (ICD) [6] and Depression, Anxiety and Stress Scale-21 (DASS-21) [7] are used to diagnose the severity of depression.

Though psychological and pharmacological treatments exist for moderate and severe depression, low- and middle-income countries are unable to provide suitable treatments and this leads to high suicidality rates [8]. As a result, 76–85% of people from those countries are suffering from depression and try to hide their symptoms of depression from others [9]. In terms of demographic factors such as gender, studies have proven more women are affected by depression compared to men [10,11]. Depression has been reported approximately 1.5 times more in women than in men [8]. Further, lots of males with depression refuse to take clinical advice from a professional consultant due to the negative attitudes towards mental health services [11]. Consequently, most males are addicted to various substances, alcohol or drugs. Sometimes, they tend to end with deliberate self-harm behaviours as a temporary solution for depression [11]. According to clinical studies, single or multiple factors such as sudden death of loved ones, unemployment, continuous exposure to violence and genetic factors cause depression.

However, there are significant challenges of using a standardised measure in diagnosing depression. For instance, the standard measures may not be validated across different populations and may lack cultural sensitivity which makes the outcome less reliable. Hence, it is crucial to take these into consideration to identify depression early and provide appropriate treatment. Though psychological experts use manual methods to identify depression, the population is generally reluctant to visit and get treatment physically by face-to-face interviews and using scoring methods [12]. Due to the existing stigma, people tend to seek solutions or, at least, would like to express their problems in online social media platforms. This study will directly benefit the stigmatised people by identifying hidden textual cues in online social media platforms related to depression and to provide support and guidance to overcome depression.

A recent survey has proven the growth of internet usage as a source of health information and 76.9% of internet users seek health-related information from the internet [13]. According to previous studies, victims of mental disorders have more tendency to gather information about their condition through the internet due to the presence of social stigma [14]. Social media and online peer communities act as digital platforms that provide information, experiential advice, and support [15] to the people with depression. Moreover, numerous studies have been conducted based on social media platforms to identify depression and extracted emotions and feelings expressed by people [16–18].

NLP is an emerging technology that uses machines to understand human languages. Although people use English as the primary language in OSF to express their problems, opinions, and guidance, they switch to other languages such as their mother tongue. By using different NLP techniques, hidden textual cues in language can be extracted and identify behavioural patterns of people. Different text mining methods such as Name Entity Recognition, sentiment analysing, relation extraction was employed in computer-based NLP applications. Exponential growth of data leads to the conducting of research based on ML. With the evaluation of ML, text mining approaches have been combined with ML in order to obtain accurate results.

The existing studies covered a broad range of computational techniques used in Depression Identification. However, given the increasing popularity of using OSF to obtain support and guidance for depression, we believe our review is appropriate on the use of computational approaches. The aim of this review is to systematically analyse the effectiveness of using NLP and ML techniques to extract information from OSF, for people with episodes of depression.

In this paper, we review how different NLP features have been extracted from textual cues posted online and how they are integrated with ML techniques in identifying depression. We are confident that this review can benefit researchers conducting experiments with social media text data for health informatics in general, and especially for computational techniques in depression identification. In summary, the main contributions of our review are:

- To review the existing problems and solutions in Depression Identification in OSF using computational techniques and comparison between them.

- To assess the strengths and limitations of existing techniques and discussion of emerging computational techniques for online social media text analytics.
- To discuss about the main open issues faced by the research community when dealing with health-related textual information and the computational approaches available.

The rest of this paper is structured as follows. The methodology section presents the papers reviewed that focused on how depression disorder is discussed on OSF, the distribution of textual cues in OSF and how it impacts society. Then, in the result section we present the state-of-the-art NLP and ML techniques used for Depression Identification and compare them. Finally in the discussion section, we include a list of open issues that outline promising directions for future research in this domain.

## 2. Methods

A Systematic review of the literature was conducted using the guidelines of the Preferred Reporting Items for Systematic Reviews and Meta-Analyses (PRISMA) in order to ensure the comprehensiveness of this effort [19].

### 2.1. Search Strategy

A systematic search of this review was conducted to assess the recent studies about the context of identification of depression-related textual clues from online digital platforms and computational approaches used to classify depression. The search was conducted electronically to choose papers published between 2010 to 2021, during October 2020 to September 2021 in the following digital libraries: PubMed, IEEE Xplore Digital Library, ScienceDirect, and these databases were chosen to cover medical and computer science aspects. To make our search strategy inclusive, at least as inclusive as possible, we explored clinical Subject headings (US National Library of Medicine) with respect to the key words. Below are keywords and searching strategies used to filter the specific articles relevant to our study. We have also excluded studies relating to the current COVID-19 pandemic, as they are still unfolding.

"Computational approach" OR "natural language processing" OR "NLP" OR "text mining" OR "text classification" OR "document classification" OR "machine learning" OR "supervised machine learning" OR "unsupervised machine learning" OR "deep learning" AND "depression" OR "data collection" OR "health care surveys" OR "algorithm" OR "evaluation" AND "depression identification" OR "depressed users" OR "social media" OR "open support forums" OR "forums".

Moreover, to examine additional articles that our search strategies on data repositories might have excluded, we manually searched for the relevant literature. The Computational Linguistics and Clinical Psychology Workshops (CLPsych) and CLEF eRisk Workshop are two leading workshops that extracted social media data to identify different sociological factors. Each year significant research has been conducted based on social media data and psychological domain areas in these efforts [12,20,21].

### 2.2. Study Selection Criteria

For this review, the following inclusion criteria must be satisfied. Our study (1) should mainly focus on depression disorder (2) should employ computational analysis to identify depression-related textual clues (3) should be based on OSF. (4) should be an original study. Since this review paper, is based on depression disorder and forum data collected in the English language, the following exclusion criteria was followed: (1) studies based on multilingual text were excluded (2) studies based on other psychological illnesses and related review papers were excluded from this study.

Due to the rapid change in internet-based technology, we have selected articles through two steps. As the first step, title, year of publication, and an abstract for each article were considered and we have restricted our search to articles published in the last 10 years. The second step excluded the duplicate articles and the articles that were not published in English. Two reviewers assessed each publication according to inclusion and exclusion

criteria independently. Figure 1 illustrates the PRISMA flowchart representing the study selection criteria and reasons for exclusion. Additionally, the PRISMA checklist is attached as a Supplementary Document.

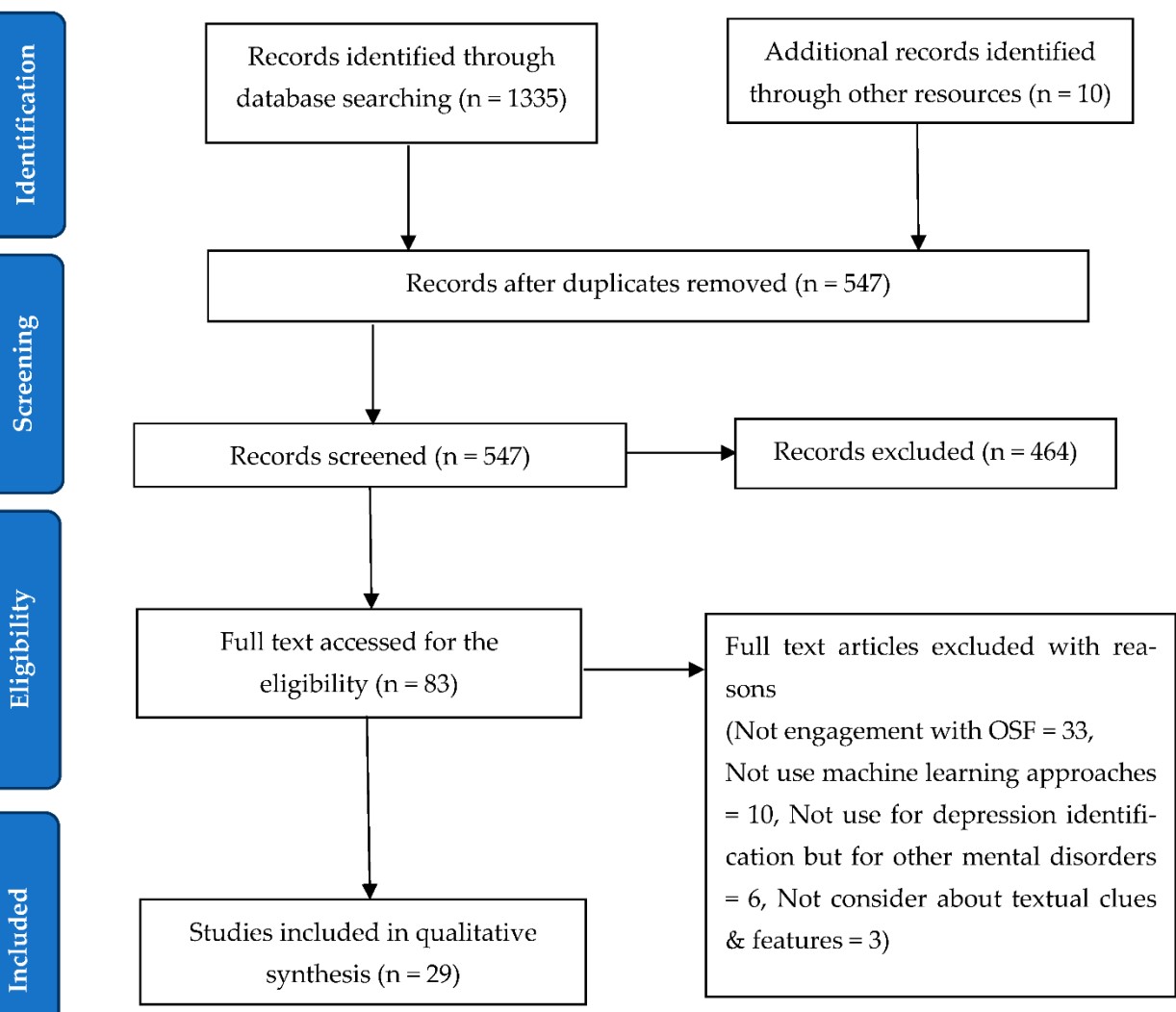

**Figure 1.** PRISMA Flowchart Diagram.

*2.3. Data Extraction*

After conducting exploratory screening of articles and obtaining relevant studies that met our inclusion criteria, we selected the most matching data from the main texts. The following data were extracted from each study: publication year, study design, publication year, sample size, study purpose, type of online digital platform (Ex: Facebook, Twitter, Reddit), target population (Ex: gender, age, sexual orientation, and so forth) and computation approaches (Ex: ML, NLP techniques). An inductive approach was taken for data abstraction based on qualitative data collected.

**3. Results**

In total, 1335 references were identified from the different databases and 10 from additional sources. A total of 547 of references were removed due to the duplication. Each reference was assessed according to the title and abstract of the study and 464 publications were excluded because they did not meet the selection criteria. After reading 83 eligible full text publications, 53 were excluded and 29 studies were selected for final review according

to the aforementioned eligibility criteria. Figure 1 presents the selection criteria of the studies.

The selected studies were categorised into several categories according to Figure 2 and most of the studies have used datasets from social media platforms to identify depression. From those studies, the majority belongs to data collected from social media such as Facebook and Twitter. Moreover, studies based on OSF such as ReachOut and Reddit were considered in this review for depression identification. Different computation approaches have been employed on behalf of depression identification. We mainly focused on NLP techniques and ML algorithms that used to perform feature extraction and depression classification.

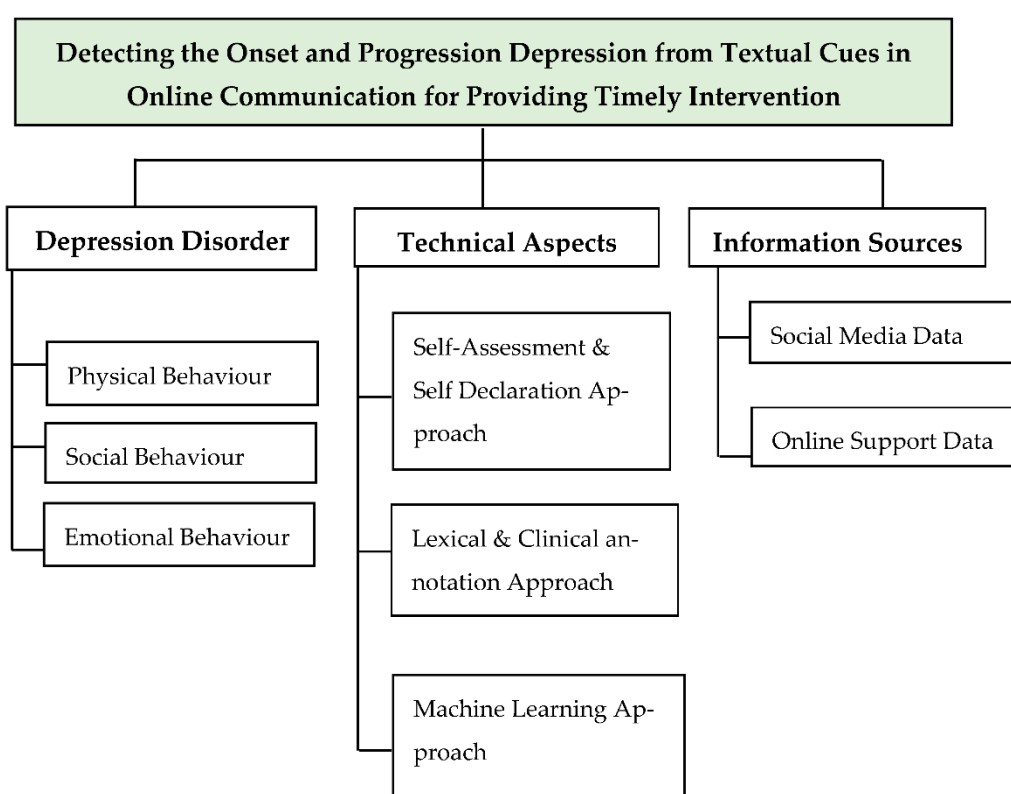

**Figure 2.** Distribution of Selected Domain areas.

*3.1. How Are Online Mental Support Forums Getting Involved with Depression Disorder?*

According to statistical reports in 2020, 4.66 billion people around the world have access to the internet and try to seek information for different purposes [22]. Since people are becoming familiar with technology more and more organisations and voluntary services create communities and connections to facilitate information through digital platforms. With technological advancement, online digital platforms such as social media and different types of community support forums act as centralised hubs that share human views, opinions and experiences with each other. This advancement of technology creates a golden opportunity for people with different mental disabilities to find solutions for their problems. Due to the mental health stigma, most people with psychological problems refuse to present themselves physically before consultants to get advice [23,24]. Based on the literature, many types of research have been conducted based on Facebook [9,25–30], Twitter [18,31–34] and Reddit [35–39] for identification of depression from textual content. When it comes to social media platforms and OSF, there is a greater propensity to discuss psychological issues on specific OSF, because people use social media platforms to share information about daily lives, not just about specific mental disorders. Furthermore, there is no specialised audience on social media platforms dedicated to providing support and guidance for psychological illnesses. However, skilled caregivers, psychiatric professionals,

clinicians, and volunteer services offer more specific clinical advice and guidance for overcoming psychological difficulties in OSF. Due to the convenience and anonymity offered by such forums, most people prefer to discuss their psychological problems with them. Moreover, detecting health problems through the OSF is becoming popular for other disorders such as cancer [40] and heart disease [41]. In the case of cancer and heart disease, there are biological markers that you can easily detect to diagnose the disease but it is quite complicated when it comes to psychological disorders because they are subjective. Based on the above factors, we hypothesize that studies that use OSF data to assess psychological illnesses are more reliable; therefore, we have considered those studies for our analysis.

OSFs are used to share advice and information related to healthcare [42] and people tend to interact with them more nowadays than in earlier years. Frequently, textual messages are used as the primary communication method between peer users in OSFs. In OSF, people are comfortable in expressing their problems in a relaxed and open manner. Additionally, when they try to communicate in OSF, they may have a comprehensive or limited understanding about the psychological condition. Causes and current symptoms of depression differ from person to person, and the way they try to express their emotions, experiences and problems are different in forms on the OSF [1]. Considering the unstructured nature of textual representations on the OSF, peers on the OSF can mislead due to the ambiguities that exist when understanding the meaning of the questions posted by a depressed person, which reduces the reliability of the answers provided by other peer members.

The actual point of posting a question on the OSF is to get a quick, reliable and accurate reply to overcome the mental health problems. Therefore, it is important to consider every factor posted on OSF and provide the most optimised answer for the person who seeks support and helps to prevent deliberate self-harm or even suicide. When considering the above factors, we have found places that need to be highlighted from both the computer science and mental health domain. Additionally, we have analysed the necessity for a solution to understand the psychological condition of a particular person and provide the best solution to overcome it based on OSF data.

Fewer studies have been conducted for support forums when compared with studies conducted for social media. One such research account is based on the ReachOut forum, which was used for detecting depressed users in OSF [36,43]. Yet they have not focused on the severity and early predictions of depression in OSF. Furthermore, some studies related to depression have been conducted based on the Reddit forum [37,44–48]. Researchers have chosen Reddit forum because it allows long posts without any limitations of words [49]. After characterising the closer connections between language and depression, some researchers are only focused on suicide-indicative and non-suicidal posts [38].

### 3.2. How Have Depression-Related Textual Clues Been Extracted from Forum Data?

The psychological experts employ different standard psychological assessments such as BDI [50], DASS-21 [51], DSM-V [29,48], Patient Health Questionnaire (PHQ) [52] to diagnose depression. Self-assessment questionnaires are used to gather information about the individual with a psychological disorder. Primarily, the self-assessment surveys assess emotional, cognitive and physical behaviours. DASS-21 scale was used to extract depression-related linguistic characteristics from user-written cover letters, letters from holidays, complaints and letters of apology [51]. Furthermore, Jana [51] tried to predict higher emotional states of depression but the study was limited as it was conducted on a quota selected sample of Czech native speakers. Some studies have used BDI assessing methodology to analyse the depressed users from Twitter posts [25]. According to the study, cognitive vulnerability and depressive symptoms have increased in the participant who had the tweets with a "past focus".

Besides the self-assessment method, self-declaration is another method used to diagnose mental disorders from public platforms such as Twitter, Facebook and OSFs. The self-declaration method allows users to declare they are already diagnosed with depression or suffered from depression in the past [53]. The CLPsych 2015 workshop [54] also investigated and researched the self-declaration posts on Twitter, related to depression. Another study was conducted based on Twitter and followed self-reported statements of diagnosis [54]. Additionally, Nguyen [55] conducted research based on forums with respect to the user's affiliation, which indicated the status of the mental condition. The main objective of the study [55] was to identify common characteristics between clinical and non-clinical forums related to mental disorders. Table 1 analyse the about the datasets, types of extracted features and NLP methods employ to extract feature.

**Table 1.** Summary of the datasets, feature types and NLP methods used in previous studies (Bag Of Words = BOW, Term Frequency/Inverse Term Frequency = TF-IDF, Part Of Speech = POS, Linguistic Inquiry Word Count = LIWC, Latent Semantic Analysis = LSA, Latent Dirichlet Analysis = LDA, Word Embedding = WD, Global Vectors Embedding = Glove, fastText Embedding = fastText, Word to Vector Embedding = Word2Vec, type-token ratio = TTR, Brunet's Index = BI, Honore's Statistics = HS).

| Dataset | Authors and Year | Post/User Counts | Types of Features | NLP Methods |
|---------|------------------|------------------|-------------------|-------------|
| CLPsych 2016 | Arman Cohan, Sydney Young and Nazli Goharian (2016) [43] | 1188 posts | Linguistic, Contextual Features, Textual Statistics | LIWC, BOW, N-gram, LDA |
| CLPsych 2017 | Anu Shrestha, Francesca Spezzano (2019) [12] | 147,619 posts | Linguistic Features, Punctuation features, Summary features, Network Features, Reciprocity, Clustering Coefficient, post embedding | TF-IDF, LIWC, WD |
| CLEF eRisk 2017 | Esteban A. Ríssola, David E. Losada, Fabio Crestani (2019) [21] | 531,453 forum posts | Semantic Proximity | LSA |
| | Alan A. Faras-Anzaldua, Manuel Montes-Gomez, A. Pastor Lopez-Monroy, Luis C. Gonz_alez-Gurrola (2017) [56] | | Post level features, User Level features | N-grams, BOW |
| | Chenlu Meng (2020) [57] | | Linguistic features | BOW, TF-IDF, WD, LIWC |
| | Maxim Stankevich, Andrey Latyshev, Evgenia Kuminskaya, Ivan Smirnov, and Oleg Grigoriev (2018) [58] | | Linguistic features, Stylometric features (text length, lexicon size), Morphological features | TF-IDF, GloVe, N gram, POS tags |
| | Marcel Trotzek, Sven Koitka, and Christoph M. Friedrich (2018) [59] | | Word and Grammar Usage, Readability, Emotions and Sentiment, Metadata Feature Summary | LIWC, GloVe, fastText |
| | Guozheng Rao,Yue Zhang, Li Zhang, Qing Cong And Zhiyong Feng (2020) [60] | | Post-level operation, User-level operation | BOW, TFIDF, LIWC, WD |
| | Sergio G. Burdissoa,b, Marcelo Errecaldea, Manuel Montes-y-G'omezc (2019) [61] | | Linguistic features | WD |
| CLEF eRisk 2018 | Sayanta Paul, Jandhyala Sree Kalyani? and Tanmay Basu (2018) [62] | 1,076,582 posts | Linguistic features | BOW, UML |
| CLEF eRisk 2019 | Sergio G. Burdissoa,b, Marcelo Errecaldea, Manuel Montes-y-G'omezc (2019) [61] | 531,453 posts | Word polarity, Mutual information, Semantic Similarity | LIWC, GloVe |
| | Faisal Muhammad Shah, Farzad Ahmed, Sajib Kumar Saha Joy, Sifat Ahmed, Samir Sadek, Md. Hasanul Kabir (2020) [63] | | word embedding techniques, metadata features | GloVe, fastText, Word2Vec |
| CLEF eRisk 2020 | Amina Madani, Fatima Boumahdi, Anfel Boukenaoui, Mohamed Chaouki Kritli, And Hamza Hentabli (2020) [35] | 35,562 posts | Linguistic features | WD |
| CLEF eRisk 2021 | Diego Maupomé, Maxime D. Armstrong, Fanny Rancourt, Thomas Soulas and Marie-Jean Meurs (2021) [64] | 2348 users | Topic Modelling, Authorship decision | WD |

**Table 1.** *Cont.*

| Dataset | Authors and Year | Post/User Counts | Types of Features | NLP Methods |
|---|---|---|---|---|
| Reddit dataset | Minjoo Yoo, Sangwon Lee, Taehyun Ha (2018) [44] | 5409 posts | Word class, emotions, Speech features | LIWC, TF-IDF |
| | JT Wolohan, Misato Hiraga, Atreyee Mukherjee, Zeeshan Ali Sayyed (2018) [45] | 23,583 posts | Linguistic features, Sentiment features | LIWC, N gram, TF-IDF |
| | Michael M. Tadesse, Hongfei Lin, Bo Xu, And Liang Yang (2019) [38] | 10,000 posts | Linguistic features | LIWC, LDA, N gram |
| | Fidel Cacheda, Diego Fernandez, Francisco J Novoa, Victor Carneiro (2019) [47] | 500,000 posts | Semantic Similarity Features, Writing Features, Textual Similarity Features, Subject Behaviour | LSA, LIWC |
| | Inna Pirina, Çağrı Çöltekin (2018) [37] | 10,000 posts | Linguistic features | N-gram |
| | Manas Gaur, Ugur Kursuncu Amanuel Alambo Jyotishman Pathak (2018) [48] | 8043 users | Statistical Characteristics, Topical Analysis, Entropy Analysis | WD, TF-IDF |
| | Amrat Mali, RR Sedamkar (2020) [65] | 13,321 posts | Psycholinguistic features | TF-IDF, BOW, LIWC |
| | Molly E. Ireland, Jonathan Schler & Gilad Gecht, Kate G. Niederhoffer (2020) [66] | 303,649 posts | Linguistic features | BOW, N-gram, LIWC, POS tags |
| | Rida Zainab, Rajarathnam Chandramouli (2020) [67] | 20,000 posts | Linguistic features | BOW, TFIDF, TTR, BI, HS |
| | Alina Trifan, Rui Antunes, S'ergio Matos, and Jose Lu'ıs Oliveira (2020) [68] | 9210 users | Absolutist Words, Analysis of Lexical Categories, Self-related Speech, Posts Length | TF-IDF, WD |
| | Iram Fatima Burhan Ud Din Abbasi Sharifullah Khan Majed Al-Saeed Hafiz Farooq Ahmad Rafia Mumtaz (2019) [69] | 1588 posts | Linguistic features | LIWC |
| | Raymond Chionga, Gregorius Satia Budhi, Sandeep Dhakal and Fabian Chiong (2021) [70] | 50,000 posts | Linguistic features | BOW, POS tags |
| MedHelp dataset | Thomas Zhang, Jason H.D. Cho, Chengxiang Zhai (2014) [71] | 1200 posts | Word Features, Pattern Features, | TF-IDF, POS tags |
| eDisease dataset | Jorge Carrillo-de-Albornoz, Ahmet Aker, Emina Kurtic, Laura PlazaID (2019) [72] | 1029 posts | Lexical, Syntactic, Network-based, Sentiment-based, Semantic feature | TF-IDF, BOW, Word2Vec |
| Online forum dataset (www. mentalhealthforum. net, www. psychforums.com, www.beyondblue.org) (accessed on 7 October 2021) | Minna Lyons, Nazli Deniz Aksayli, & Gayle Brewer [16] | 463 Posts | Linguistic analysis | LIWC |

When considering recent studies, extraction of information from OSFs has been popular throughout the globe and many people interact with technology to find solutions for their psychological problems. Since forums represent information as textual formats, different techniques have been employed to extract essential features. A lexical, self-annotated approach was one way of information extraction. Specific depression-based lexical resources such as Lexical Depression-oriented dictionaries have been employed to analyse the level of depression in texts automatically [21]. Losada [21] developed a corpus that contains texts written by depressed and non-depressed users and the level of depression was estimated according to a ranking procedure based on three search strategies (Lemma-PoS, word embedding and Explicit context-based study). Another study which was conducted in 2019 has also focused on Lexical features such as Bag of Words (BOW) and Noun Phrased to extract facts, opinions and experiences from health forums [72]. Another study based on Reddit forum has identified a lexicon of terms that are common among depressed users by analysing the frequency of N-gram features, Latent Dirichlet Allocation features and LIWC (Linguistic Inquiry and Word count) features [46]. By considering that stigma exists in depressed users, another study was conducted using LIWC features to detect linguistic traces of depression [45]. Figure 3 analyses the textual features used on the selected studies.

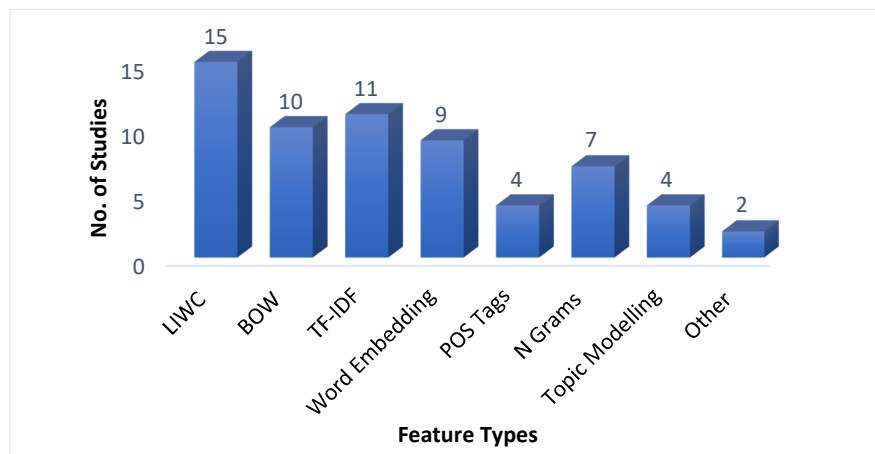

**Figure 3.** Analysis of textual features used on selected studies.

The annotation process was carried out manually according to the keywords related to mental issues after information extracted from public communities such as Twitter [73], Reddit [39]. Most of the time annotators followed assessment manuals such as DSM-V [74] to identify symptoms related to depression. The process was conducted by reputed annotators or by using a bottom-up classification method based on predefined theories. Other than keywords-based annotation, some studies related to depression have focused on the personal experiences shared by users on the forums. Although the manual annotation of posts is labour-intensive, factors such as ethnic groups and weather infer hidden personal experiences related to psychological issues. Though most studies have focused on the self-assessment questionnaire method due to the reliability of data for prediction, it takes additional costs. Moreover, when reviewing the literature, we found less research based on forum depression mining and none focused on monitoring the severity level of depression, suggesting most optimised similar existing solutions on the forum. We were influenced by this finding to create our own corpus extracted from public mental open support forums and predict via a hybrid approach on data that have been annotated automatically.

*3.3. What Kind of Machine Learning Approaches Have Been Considered in Order to Classify Depression Disorder from Forum Data?*

Researchers conducted mainly supervised learning-based approaches after annotating data to depressed or not depressed labels. Classification algorithms such as Support Vector Machines (SVM), Logistic Regression (LR), and Random Forest (RF) are frequently used to classify depressive posts from OSF. Moreover, some studies have also used topic modelling and clustering algorithms, both of which are unsupervised learning approaches, to identify depression in textual data. Deep learning is a subfield of machine learning that enables various types of research to be conducted in order to identify depression. By adapting transformer-based language model architectures such as Bi-directional Encoder Representation Transformer (BERT), and Generative Pre-Trained Transformer (GPT), researchers were able to improve the final accuracy of the models. To evaluate the ML algorithms, most of the studies used the following methods combining different features.

Accuracy is the proportion of true results (both true positives and true negatives) among the total number of cases examined, and is computed as follows:

$$\text{Accuracy} = \frac{\text{true positive} + \text{true negative}}{\text{true positive} + \text{true negative} + \text{false positive} + \text{false negative}}$$

F-measure is the harmonic mean of precision and recall, and is computed as follows:

$$\text{F measure} = \frac{2 \times \text{recall} \times \text{precision}}{\text{recall} + \text{precision}}$$

where precision is defined as:

$$\text{Precision} = \frac{\text{true positive}}{\text{true positive} + \text{false positive}}$$

and recall is:

$$\text{Recall} = \frac{\text{true positive}}{\text{true positive} + \text{false negative}}$$

Table 2 illustrates findings and computational approaches use in selected studies independently.

**Table 2.** Matrix of the summary of findings in depression identification using online support forum data; Moreover similar techniques were applied when analysing online forum textual contexts (Support Vector Machines = SVM, Logistic Regression = LR, Random Forest = RF, Ada Boosting = AB, Multinomial Naive Bayes = MNB, K-nearest neighbor = KNN, Passive Aggressive Classifier = PAC, Stochastic Gradient Descent = SGD, Neural Network = LSTM, Multi-layer Perceptron = MLP, Conventional Neural Network = CNN, Long Short Term Memory = LSTM, Gradient Boosting = GB, Bagging Predictors = BP, Bidirectional Long Short-Term Memory = BiLSTM, Multi-Gated LeakyReLU CNN = MGL-CNN, Bi-directional Encoder Representation Transformer = BERT, Generative Pretrained Transformer = GPT).

| Author & Year | Dataset (Type) | Algorithms Used | Findings |
|---|---|---|---|
| Arman Cohan, Sydney Young and Nazli Goharian (2016) [43] | CLPsych 2016 | SVM, RF, AB, LR | Model for identifying depression and self-harm posts on ReachOut forum based on Lexical, contextual, Topic and statistical features. |
| Esteban A. Ríssola, David E. Losada, Fabio Crestani (2019) [21] | CLEF eRisk 2017 | SVM | Model to measure the semantic proximity between textual posts and a set of words with topical relevance to depression. |
| Minjoo Yoo, Sangwon Lee, Taehyun Ha (2018) [44] | Reddit data | Semantic network analysis | Word usage of posts and comments in online communities for bipolar and depressive disorder to understand how people perceived these mental disorders and shared their experiences. |
| JT Wolohan, Misato Hiraga, Atreyee Mukherjee, Zeeshan Ali Sayyed (2018) [45] | Reddit | SVM | Linguistic patterns of depressed Reddit users are consistent with popular depression batteries. |
| Michael M. Tadesse, Hongfei Lin, Bo Xu, And Liang Yang (2019) [20] | Reddit | SVM, MLP | Model to detect any factors that may reveal the depression attitudes of relevant online users in Reddit users' posts. |
| Fidel Cacheda, Diego Fernandez, Francisco J Novoa, Victor Carneiro (2019) [47] | Reddit | RF | Model to detect depressed subjects and nondepressed subjects based on features defined from textual, semantic, and writing similarities. |
| Thomas Zhang, Jason H.D. Cho, Chengxiang Zhai (2014) [71] | MedHelp | SVM | Machine learning approach to identifying user intents from original thread posts from online health forums related to depression. |
| Anu Shrestha, Francesca Spezzano (2019) [12] | CLPsych 2017 | KNN, LR | Model to detect depressed users in online forums using network features and linguistic features. |
| Jorge Carrillo-de-Albornoz, Ahmet Aker, Emina Kurtic, Laura PlazaID (2019) [72] | eDiseases dataset | SVM | Automatic classifier to predict patient-generated contents based on lexical, syntactic, semantic, network-based and emotional properties of texts related to depression. |
| Inna Pirina, Çağrı Çöltekin (2018) [37] | Reddit | SVM | Identify sources for successful detection of depression from social media text (by employing different datasets) |

**Table 2.** *Cont.*

| Author & Year | Dataset (Type) | Algorithms Used | Findings |
|---|---|---|---|
| Sayanta Paul, Jandhyala Sree Kalyani and Tanmay Basu (2018) [62] | CLEF eRisk 2018 | AB, LR, RF, SVM, RNN | Frameworks to early identify anorexia or depression of the individual documents using state-of-the-art classifiers. |
| Manas Gaur, Ugur Kursuncu Amanuel Alambo Jyotishman Pathak (2018) [48] | Reddit | RF | Algorithm on analysing subreddits to utilise the curated medical knowledge bases to quantify relationship to DSM-V categories. |
| Alan A. Faras-Anzaldua, Manuel Montes-Gomez, A. Pastor Lopez-Monroy, Luis C. Gonz_alez-Gurrola (2017) [56] | eRisk2017 | NB | Approach to identify depression through online publications, which is based on individual posts to characterise users' behaviour and analyse users from a higher point of view. |
| Diego Maupomé, Maxime D. Armstrong, Fanny Rancourt, Thomas Soulas and Marie-Jean Meurs (2021) [64] | CLEF eRisk 2021 | NN | Task 3 was designed to measure the Severity of the Signs of Depression through BDI scores based on similarity measurement. |
| Amrat Mali, RR Sedamkar (2020) [65] | Reddit | NB, KNN | Building a topic model to identify hidden topics that act as a depression triggering points. |
| Minna Lyons, Nazli Deniz Aksayli, & Gayle Brewer [16] | Online forums | Statistical Analysis | Model to identify distinctive linguistic patterns displayed by those experiencing mental health and difficulties interactive online communication |
| Molly E. Ireland, Jonathan Schler & Gilad Gecht, Kate G. Niederhoffer (2020) [66] | Reddit | CNN | Implemented an array of machine learning and regression models to classify posts from neutral Reddit forums as written by depressed users (self-identified) or random controls. |
| Amina Madani, Fatima Boumahdi, Anfel Boukenaoui, Mohamed Chaouki Kritli, And Hamza Hentabli (2020) [35] | eRisk 2020 | CNN, Bi-LSTM | Model to measure the severity of the signs of depression from a thread of user post. |
| Sergio G. Burdisso, Marcelo Errecalde, and Manuel Montes-y-G´omez (2021) [50] | CLEF eRisk 2019 | SVM, GPT | Machine learning model to demonstrate that language usage can provide strong evidence in detecting depressive people, connecting with the severity level of depression. |
| Chenlu Meng (2020) [57] | CLEF eRisk 2017 | LR, SVM | Supervised classifier to act as a preliminary screening method before depression diagnosis. |
| Maxim Stankevich, Andrey Latyshev, Evgenia Kuminskaya, Ivan Smirnov, and Oleg Grigoriev (2018) [58] | CLEF eRisk 2017 | SVM, CNN | Automatic detection of depression signs from textual messages of Russian social network VKontakte user |
| Marcel Trotzek, Sven Koitka, Christoph M. Friedrich(2018) [59] | CLEF eRisk 2017 | LR | Model on early detection of depression using machine learning and based on messages on a social platform. |
| Raymond Chionga, Gregorius Satia Budhi, Sandeep Dhakal and Fabian Chiong (2021) [70] | Reddit, Twitter, Victoria's diary | AB, RF, GB, BP | Generalised approaches using ML methods and social media texts can be effectively used to detect signs of depression. |
| Rida Zainab, Rajarathnam Chandramouli (2020) [67] | Reddit | LR, RF | Machine learning and Explainable AI analysis of depression and non-depression reddit text data in English and Urdu language. |
| Faisal Muhammad Shah, Farzad Ahmed, Sajib Kumar Saha Joy, Sifat Ahmed, Samir Sadek, Md. Hasanul Kabir (2020) [63] | CLEF eRisk 2019 | Bi-LSTM | Model to classify depressed users but also to reduce the amount of time to predict the state of the users |

<div align="center">Table 2. <em>Cont.</em></div>

| Author & Year | Dataset (Type) | Algorithms Used | Findings |
|---|---|---|---|
| Guozheng Rao,Yue Zhang, Li Zhang, Qing Cong And Zhiyong Feng (2020) [60] | Reddit and CLEF eRisk 2017 | MGL CNN, SVM, MNB, LSTM | Posts representations models for identifying depressed individuals, which was more accurate and efficient than general early depression detection models. |
| Alina Trifan, Rui Antunes, S'ergio Matos, and Jose Lu'ıs Oliveira (2020) [68] | Reddit | SVM, PAC, MNB, SGD | A model based on hand-crafted psycholinguistic features as possible improvements to standard classification approaches of depressed online personas. |
| Iram Fatima Burhan Ud Din Abbasi Sharifullah Khan Majed Al-Saeed Hafiz Farooq Ahmad Rafia Mumtaz (2019) [69] | Reddit | MLP, SVM, LR | A model that combines the text-based features and machine learning techniques to classify depressive and non-depressive posts and then further identify posts representing characteristics of postpartum depression |
| Sergio G. Burdisso, Marcelo Errecalde, Manuel Montes-y-G'omezc (2019) [61] | CLEF eRisk2017 | KNN, LR, SVM, NB | SS3 (incremental, early classification and explain-ability), a novel text classifier that can be used as a framework to build systems for early risk detection ERD |

Supervised learning-based algorithms and text mining approaches were able to archive good results. Most of the studies conducted their experiments based on lexical, syntactic and semantic features related to depression collected from user posts. Some studies explained that the network features provide additional support when identifying depressed users from online forums. As an overall evaluation, computational models developed using supervised learning classifiers and text mining approaches were able achieve 70–90% range accuracy. Figure 4 assesses the classification algorithms used on selected studies.

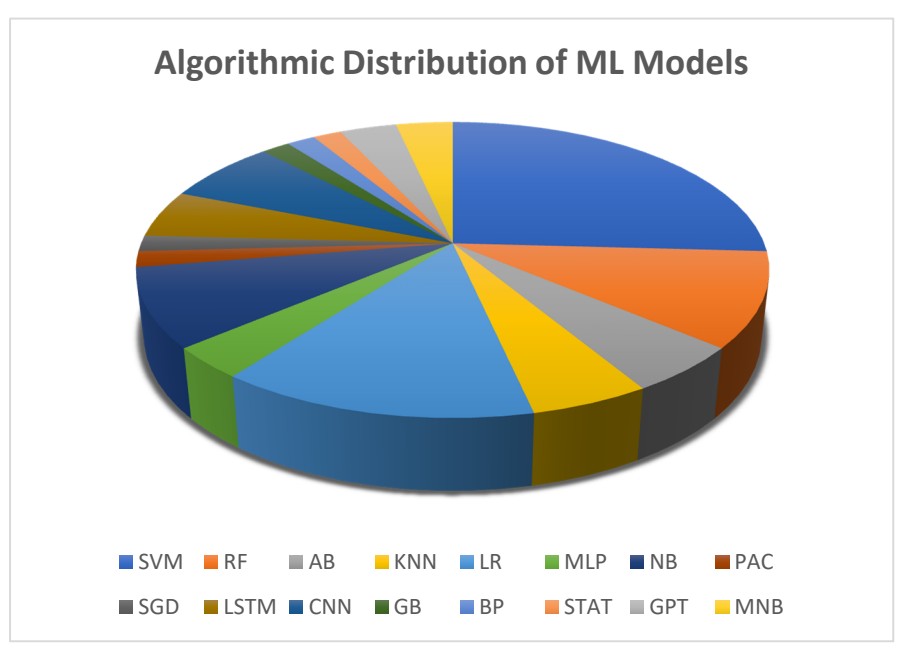

**Figure 4.** Distribution of classification algorithms on selected studies.

## 4. Discussion

The main goal of this review was to investigate how NLP and machine learning techniques have been employed to predict the identification and severity of depression disorders through OSF. This review reveals a significant shortcoming of studies that predict the progression of the severity levels in a sufferer. When considering articles collected

through our search queries and the results obtained from our review, many studies were based on social media networks such as Twitter and Facebook with a limited number related to OSF.

The majority of studies had focused on identifying depression-related posts from support forums, in which a high number were related to identifying self-harm or suicidality-related behaviour. Many of these experiments were conducted on small sets of samples, which brings into question their robustness and ability to be generalised. Considering the feature extraction process, numerous methodologies have been employed and fall into subgroups of (a) Linguistic based: LIWC features, articles, prepositions, auxiliary verbs, adverbs, conjunctions, personal pronoun, impersonal pronouns, verbs, and negations, semantics; (b) Network based: count of posts, count of replies and (c) Temporal based: emotional features. Additionally, supervised learning algorithms were the most employed technique in previous studies of depression classification based on the features extracted. The classification tasks and the knowledge extraction process can be extended based on more suitable features such as demographic and personal health information without limiting and adapting common categories associated in social media. Though multiple studies are available for depression identification, deep learning mechanisms have been rarely used with textual features.

### 4.1. Open Issues and Limitations

Some limitations that we identified in the process of depression identification and our proposed suggestions to address these limitations are given below:

- Post validation: most of the time, without having much knowledge on depression disorder, forum users are often confronted with misleading information or misdiagnose themselves. Therefore, to easily identify relevant symptoms from online forum data available as textual representations, we need a good quality analyser able to identify variations in behavioural and emotional patterns.
- Semantics-based understanding of the post: since forum posts are lengthy in nature, some consist of contradictory semantics, which misleads the extraction of the overall meaning of the post, making it hard for machines to understand depression disorder accurately.
- Severity assessments: since depression disorder has different severity levels, treatments vary accordingly requiring the need to identify unique textual characteristics for each stage. Only after identifying the degree of severity can a forum moderator or clinician provide reliable recommendations.
- Anomalous vs. normal user identification: due to social stigma, most depressed users of OSF try to be anonymous. Being able to access linked data to their other social media profiles would enable the more accurate diagnosis of their depression.
- Technological barrier: This review focuses only on the questions asked in OSF which requires by definition that only those who have access to the internet and satisfy the technological requirements are covered. There still are many people who suffer from depression who do not have any access to such technology platforms.
- Cultural and language barrier: In different cultures, different languages are used by people to communicate with each other. As a result, people try to express their emotions, symptoms of disease and sensitive information through unique terms and expressions that belong to their own language and cultures.

### 4.2. Implication of Future Research

Technological advances in recent years have led to a significant amount of data and techniques such as sentiment analysis and opinion mining to analyse what people say or share in their everyday life [75–77]. As it is, there are both good and bad aspects of technology dependence.

An Internet-based framework that anyone can access from anywhere to seek accurate information and clinical guidelines relevant to depression will be beneficial to the whole

community. The proposed study is an interdisciplinary investigation of the connection between the psychological states of digital natives and their online behaviour. As such it draws on the fields of Psychology and Computer Science to explore the extent to which public content from OSF can be a source of information to both extract cues for the early detection of psychological disorders of individual sufferers.

Recent studies show that researchers can analyse static data on a forum and predict whether it contains depressive contents or not. However, it is now time to integrate forum users with conversational AI to obtain more dynamic data in the process of diagnosing depression. The intersection of technology and mental health is an important aspect to consider in understanding depression; because more details are shared freely when anonymous, going beyond explicit self-reporting (as needed by the more common questionnaire method) to provide platforms for peer support can help with dealing with depression. Even though this may not be professional help, it could form a first line of defence in the form of a community of support. These could all be the basis for further understanding depression.

In today's competitive age, people interact with technology to seek information related to physical, psychological, and social well-being. Though it is difficult to find large-sized medical textual data, it is worthwhile to research how to extract those online data relevant to human's psychology. Additionally, with the improvement of ML, deep learning techniques can work with large size textual corpus varying with time which the other techniques cannot. Therefore, we believe timely assessment of textual features with deep learning should be investigated in the future. Deep reinforcement learning is a novel area of ML which enables an agent to learn from an interactive environment by experimenting with feedback according to its own experience. So, if we combine reinforcement learning with deep learning to detect the severity of depression disorder from forum posts, then better results can be obtained.

## 5. Conclusions

Our main purpose in this review is to provide an overview of the state-of-the-art research on NLP and ML techniques employed in developing predictive models to recognise depression disorders from OSF. Furthermore, this study analysed the usefulness of the textual clues that exist on open support forums for depression. As major concerns, this review shows that only a few studies were conducted for addressing how emotional, physical and social behaviours affect human psychological well-being. Moreover, from the above section, we introduced some future directions that our research can help to formulate and validate future novel classification models for recommending individually tailored interventions based on the severity level of depression disorder in real-time for the users engaged in online support forums. However, the reliability of the data and the general desirability of such processes should be carefully analysed and be respectful of users.

**Supplementary Materials:** The following are available online at https://www.mdpi.com/article/10.3390/info12110444/s1, Table S1: PRISMA Checklist.

**Author Contributions:** Conceptualization, I.A.N.A., P.S. and R.W.; methodology, I.A.N.A. and R.W.; software, I.A.N.A.; validation, I.A.N.A. and P.S.; formal analysis, I.A.N.A., P.S. and R.W.; investigation, I.A.N.A., P.S. and R.W.; resources, I.A.N.A. and P.S.; data curation, I.A.N.A. and P.S.; writing—original draft preparation, I.A.N.A., P.S. and R.W.; writing—review and editing, I.A.N.A., P.S. and R.W.; visualization, I.A.N.A., P.S. and R.W.; supervision, R.W.; project administration, R.W. All authors have read and agreed to the published version of the manuscript.

**Funding:** This research received funding from University of Colombo School of computing, Sri Lanka.

**Institutional Review Board Statement:** Not applicable.

**Informed Consent Statement:** Not applicable.

**Data Availability Statement:** The datasets used for this study is available on this GitHub repository: https://github.com/isuri97/Depression_SocialMedia_Datasets (accessed on 14 October 2021).

**Acknowledgments:** We would like to acknowledge Vincent Halahakone for correcting English on our manuscript and thank all who helped in various ways to make this work bear fruit.

**Conflicts of Interest:** The authors declare no conflict of interest.

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
