# Peer review of "Investigating Machine Learning & Natural Language Processing Techniques Applied for Predicting Depression Disorder from Online Support Forums: A Systematic Literature Review"

_information, doi:10.3390/info12110444_

Round 1

Reviewer 1 Report

Review of

Investigating Machine Learning & Natural Language Processing techniques applied for predicting depression disorder from Online Support Forums: A Systematic Review

This systemic review includes articles that use online support data to predict/categorise depression. The manuscript is very interesting, timely, and informative. However, unfortunately I see some issues with how the paper is today, which would probably require extensive work. Good luck with this important work. Please see my more detailed comments below.

In regards to the first sentence in the abstract: ”Depression is a common mental health disorder which affects an individual’s mood”; please note that depression affects more than just the mood, including for example cognitions (thoughts patterns), sleep, etc. Although the core criteria include low mood and loss of interest according to the DSM-5 framework.

The beginning of the introduction I think needs references (and perhaps could be removed because it is not very central for the article):“After the industrialization era, people are focusing more on money as their priority while keeping the quality of their lives secondary. This is one of the reasons for how mental disorders have become more prevalent among the population.”

When you discuss depression throughout the paper, please be clearer about what you mean; for example, are you always referring to Major Depressive Disorder?

Considering discussing the difficulty of diagnosing depression, i.e., getting a gold standard. For example, what is the difference of using rating scales, unstructured/structured interview (such as the M.I.N.I.). Using gold standard to predict depression is very rare because it is very expensive and resource intensive.

One article that I think is particularly good is:

Eichstaedt, J. C., Smith, R. J., Merchant, R. M., Ungar, L. H., Crutchley, P., PreoÅ£iuc-Pietro, D., ... & Schwartz, H. A. (2018). Facebook language predicts depression in medical records. Proceedings of the National Academy of Sciences115(44), 11203-11208, which predict MDD from medical records (whilst acknowledging that this is not the gold standard as well).

What it the difference between: depression identification versus diagnosing depression.

How can clinicians practically use it when many are anonymous?

Perhaps consider discussing why you narrowed it down to only include social support forums; and how is it different from other social media data such as Facebook status updates.

In terms of ”gender studies have proven”, consider not using such as definitive language. Studies from social sciences often provide ”supporting evidence”, rather than “proving” something (which is more common in, for example, math).

Please consider referencing: “Further, lots of males with depression are refusing to take clinical advice from a professional consultant.”

Please consider elaborate and more clearly describe these aspects: “Symptoms of depression are categorised into three stages 1) Mild  2) Moderate and 3) Severe, Standard measures such as Beck’s Depression Inventory (BDI) [4], DSM-V[5], ICD [6] and DASS-21 [7] are used to diagnose the severity of depression. ”

Although the introduction is interesting, some parts I think simplifies things and misses what I consider one of the most important aspect, namely: how depression is being defined and then measured in research/clinical settings. 

Please consider referencing: Moreover, numerous studies have been conducted based on social media platforms to identify depression and extracted emotions and feelings expressed by people.

Not sure what “social data” is, do you mean something like social media text data?

You write: “Moreover, to examine additional articles that our search strategies on data repositories might have excluded, we manually searched for the relevant literature on the proceedings of the Computational Linguistics and Clinical Psychology Workshops (CLPsych) and CLEF 2021 Workshop (ERISK).”. So did you find any new articles this way? Because in that case that could indicate that your previous search criteria were not broad enough and if you didn’t find many new articles it could indicate your criteria were good.

Please consider going over the English in more detail. At times I had difficulties understanding everything fully.

How was it decided it was “mainly focused on depression disorder” and why?

In regards to: “data collected from other languages”; other languages than what? And why?

You say that “empirical studies were excluded” , but why did you exclude them?

I’d like to see this supported my research: “Due to the mental health stigma, most of the sufferers refuse to get direct advice from physical consultants”

The references in this sentence seam a bit problematic to me: Moreover, detecting health problems through the online forum is becoming popular for other disorders such as cancer [25] and heart disease [26]. Reference 25 appears to rather be about personalized information retrieval / about getting appropriate information about a specific problem, whereas reference 26 appears to be about analyzing Electrocardiogram (ECG)?

Some parts I find a little bit difficult to understand: for example, “furthermore, an individual’s mental severity levels, causes and current conditions are different from person to person and try to express the emotions, experiences and problems in different forms on the online support forms [1]”.

And

”Although the manual annotation of posts is labour-intensive, hidden factors related to mental health issues can be inferred such as ethnic groups, weather.”

Please consider referencing: “Clinicians follow standard psychological assessments to diagnose depression”. And what do you really mean by ”standard psychological assessments”. Different clinicians tend to do very different things when diagnosing depression (unfortunately).

What do you mean by self-declaration here: “Besides the self-assessment method, self-declaration is another method used to diagnose mental disorders from public platforms such as Twitter, Facebook and forums.”

Table 1 could preferably be presented earlier, I think. I would also suggest to including more information in the table, such as indicating how depression was defined and assessed, the predictive accuracy of the depression models, separating between NLP and ML algorithms, the N participants, and whether dataset and code are open.

According to the Findings column many articles appears to be about self-harm or suicide rather than depression; and this is not the same thing.

This finding: “Model social media images to study relation be- tween depression and anxiety to the content of im- ages that people post or based on the profile pictures on social media” appears to be about profile pictures rather than text in an online support forum?

Please consider if this is ethical: But if there a way to identify their true identity, other digital profiles can also be check when identifying the depression.

Perhaps you could also consider how this type of research can inform research on understanding depression/MDD from a research perspective.

Reviewer 2 Report

I suggest including the metrics in the summary,

figure 2 is not clear, I suggest modifying it,

I find the study interesting, I suggest doing the same with a large database.
